# Wearable Devices for Environmental Monitoring in the Built Environment: A Systematic Review

**DOI:** 10.3390/s21144727

**Published:** 2021-07-10

**Authors:** Francesco Salamone, Massimiliano Masullo, Sergio Sibilio

**Affiliations:** 1Construction Technologies Institute, National Research Council of Italy (ITC-CNR), Via Lombardia, 49, San Giuliano Milanese, 20098 Milano, Italy; 2Department of Architecture and Industrial Design, Università degli Studi della Campania “Luigi Vanvitelli”, Via San Lorenzo, Abazia di San Lorenzo, 81031 Aversa, Italy; massimiliano.masullo@unicampania.it (M.M.); sergio.sibilio@unicampania.it (S.S.)

**Keywords:** environmental monitoring, wearable devices, wearables, systematic review, visual environmental factor, acoustic environmental factor, thermal environmental factor, air quality environmental factor

## Abstract

The so-called Internet of Things (IoT), which is rapidly increasing the number of network-connected and interconnected objects, could have a far-reaching impact in identifying the link between human health, well-being, and environmental concerns. In line with the IoT concept, many commercial wearables have been introduced in recent years, which differ from the usual devices in that they use the term “smart” alongside the terms “watches”, “glasses”, and “jewellery”. Commercially available wearables aim to enhance smartphone functionality by enabling payment for commercial items or monitoring physical activity. However, what is the trend of scientific production about the concept of wearables regarding environmental monitoring issues? What are the main areas of interest covered by scientific production? What are the main findings and limitations of the developed solution in this field? The methodology used to answer the above questions is based on a systematic review. The data were acquired following a reproducible methodology. The main result is that, among the thermal, visual, acoustic, and air quality environmental factors, the last one is the most considered when using wearables even though in combination with some others. Another relevant finding is that of the acquired studies; in only one, the authors shared their wearables as an open-source device, and it will probably be necessary to encourage researchers to consider open-source as a means to promote scalability and proliferation of new wearables customized to cover different domains.

## 1. Introduction

The so-called Internet of Things (IoT), a spider web of networked and interconnected objects that have proliferated over the past decade, could have a far-reaching impact on determining the relationship between human health and environmental quality [1] due to its ubiquitous intelligence [2].

Various studies have made it possible to thoroughly investigate the potential of IoT to improve all aspects of our lives, from industrial IoT to connected health or smart cities [3,4,5]. Among the IoT devices, the smartphone is the most widely used. In fact, the smartphone has become a disruptive presence in all essential various human activities: in some applications, the smartphone can be used to check the health of the users [6], or it can be used to acquire geographic coordinates from the real life of the users, which define the most relevant “place-of-interest”, described as “a location where the user usually goes and stays for a while” [7]. Smartphones can also be used to record information for user-profiling, and they can also influence the level of life satisfaction [8,9].

While smartphones have been confirmed as a ubiquitous technology that inevitably interferes with our daily lives, more and more users are also interested in using wearable devices or simply wearables. Google Trends, a service that provides accurate and representative information about users’ online search habits, reports on the global rise in consumer interest over the past decade for the keyword “wearable” (Figure 1).

Figure 1 highlights the increasing interest of consumers in wearable devices, who are probably attracted by the desire to improve the quality of life or because their behaviour may also be influenced by IT vendors who, because the smartphone market is mature, try to create new advertising campaigns [10] to create a new demand for mobile devices, focusing their attention on smartbands and smartwatches [11] without considering real needs and investing in research to prove the effectiveness of their products [12].

While there are different practical experiences with the use of wearables for ergonomics [13] or health monitoring [14,15] and workers’ safety [16], few researchers have investigated the impact of IoT-based infrastructure in environmental monitoring in the last decades. In [17], the authors systematise knowledge in the field of industrial wearables’ safety to assess the relevance of their use in enterprises as the technology maintaining occupational safety, to correlate the benefits and costs of their implementation, and, to outline promising directions for future work in this area.

In [18,19], the authors performed an extensive literature review of low-cost sensing technologies for air quality monitoring and exposure assessment and for monitoring air pollution and physical activity using sensors. In [20], the authors reviewed miniaturised sensors to measure airborne gasses and Particulate Matter (PM). In [21], the potential of different sensor-based modules for air quality monitoring was analysed in terms of power consumption, cost, response time, and lifetime. In [22], the authors presented a literature review of current research and developments on wearable devices from both academia and industry to be considered for environmental monitoring as a whole. In [23], the authors focused on low-cost IEQ sensors and cloud-based platforms to create holistic, personalized, and scalable well-being monitoring systems. They emphasised the need for wearables to create personalised approaches to IEQ monitoring.

Our study differs from the above works. It is the only one that conducts a systematic review on the four different environmental factors (visual, acoustic, thermal, and air quality) and uses the results to define the trends of research of wearables for environmental monitoring.

But, when was the term wearable introduced, and how can it be defined? What are the main considered areas of environmental monitoring? The following two sections will provide answers to these questions.

### 1.1. Introduction to Wearables

Even though the first example of clothing with smart capabilities referred to the 1980s, with an example of a shoe-based computer, designed and developed to assist gamblers in a casino [24], the term “wearable” was first used in the scientific literature in 1996 when, in [25], the author presented a personal imaging system and, in [26], where the author gave an overview of energy generation during the user’s daily activities, removing the technological limitation of batteries to power wearables. In 1997, the researchers of MIT media laboratory, Picard and Healey, used the term “affective wearables” [27] to refer to a system equipped with sensors that allowed detecting affective patterns, such as heart rate variability and electrodermal activity. To date, the class of wearable electronics or technologies, called “wearables” for short, has attracted increasing public interest and is generally identified as a category of devices that can be worn or tattooed on the human skin or even implanted in the human body to continuously and accurately monitor some variables (biometric in most of the cases, but also environmental in some other cases) without interrupting or restricting the user’s movements [28].

The United International Children’s Emergency Fund (UNICEF) also addressed the issue of wearables by proposing a competition on wearable design, with the intent of highlighting the ethical implications of wearable research. In this context, UNICEF aimed to highlight the potential of wearables to not only help consumers become more active, but also to improve their quality of life, thus promoting social interest among the public. UNICEF recommended the following requirements for potential wearables: low-cost, low power consumption, robustness and durability, scalability, and created to be open source and in the public domain [29,30].

The many commercial wearables that have entered the market in recent years do not seem to meet the above prerequisites but rather serve commercial purposes by connecting watches, glasses, jewellery, which differ from standard devices only by the term “smart” [31], to the smartphone in order to improve its functionality, enabling, for example, payment for commercial objects, monitoring of physical activity [32], and collection of personal biometric data.

### 1.2. Environmental Factors to Be Monitored

Environmental Quality (EQ) can be subdivided by analysing it in terms of Indoor EQ (IEQ) or Outdoor EQ (OEQ). Both IEQ and OEQ are important to ensure the health and well-being of people.

It is well known that users spend a large part of their time indoors, so the quality of environments within buildings and the satisfaction and well-being of occupants is a hot topic today [33].

Especially in low-cost housing, where the limited indoor space may lead occupants to spend more time doing various outdoor activities, the quality of the outdoor environment is crucial [34].

Both IEQ and OEQ refer to a holistic concept that includes various environmental factors: visual, acoustic, thermal, and air quality.

Considering the thermal aspect in indoor environments, it is possible to improve occupant satisfaction and productivity [35]. The thermal aspect is also widely considered in outdoor spaces to understand the human response to the environment, especially in the hot season when the heat island effect, defined as the phenomenon in which the temperature recorded in an urban area tends to be higher than the surrounding areas, becomes more an issue to be considered by policymakers and for which mitigation measures have to be considered [36].

The air quality factor is another important aspect to be considered, which dramatically impacts the productivity and health of users indoors [37] and outdoors [38]. The concentrations of some variables or indoor pollutants (CO_2_, VOCs, PM, NO_2_) correlate strictly with those monitored outdoors, depending on the ventilation settings and indoor or outdoor production. As can be seen, air quality is a very important issue and is often analysed through wearables.

Visual environmental factor in indoor spaces has been shown to affect mental health and productivity of occupants [39]. It could have a cross-modal or combined effect on the perception of other Environmental Effects (EFs). This is probably the only EF studied more thoroughly in indoor environments due to controlling and optimising artificial lighting systems and daylighting.

The acoustic EF has been widely studied indoors and outdoors. Many of the principles and investigations could be applied indiscriminately indoors and outdoors [40,41,42,43,44].

In all cases, monitoring all four environmental factors could be helpful to understand the complex area of interaction (cross-modal or combined) among the different environmental aspects and user perception of IEQ or OEQ. In order to link the term wearables with the environmental aspect, some new acronyms have been introduced to emphasise the class of IoT devices for monitoring some environmental factors [45]: Personal Environmental Monitoring System (PEMS), an IoT device, to measure indoor environmental exposure to contaminant [45] and Wearable Environmental Monitoring System (WEMS), usually used outdoors, which can also communicate with PEMS [45].

This article aims to answer the following main question: What is the global trend in the scientific literature regarding environmental monitoring with wearables? To this aim, a customised query was used through a specific search engine.

The following sections describe the methodology to define the query and how the acquired data were used to perform the systematic review and the most relevant scientific literature findings related to wearables and environmental monitoring. Limitations are discussed in the conclusions section, with some future improvements.

## 2. Methodology

The methodology for collecting the papers consists mainly of two steps: the identification of the database to use and the definition of the query for downloading the metadata used for the systematic review.

The first was about establishing the database from which the information was obtained. In this case, the Scopus database was selected. Among the different available databases (e.g., PubMed, Web of Science, Google Scholar), the Scopus search engine, developed by Elsevier, was used. Its resources include Institute for Scientific Information (ISI) and Scopus-indexed papers. Scopus focused primarily on the field of physical sciences, health sciences, life sciences, and social sciences [46], which is consistent with the aim of this research. The covered period is from 1966 to the present [46]. Also, this aspect is consistent with our systematic review since, as reported in the previous Section 1.1, the oldest paper presenting a “wearable” device with computational capabilities was published in the 1980s. More details about the characteristics of the main databases can be found in [46]. Scopus provides an advanced search that allows more operators and codes to make a specific query. The search results in Scopus can be exported in different formats [46]. Scopus provides all metadata as furnished by publishers for all indexed content, including authors, affiliations, the title of the paper, abstract, authors and related affiliations, year of publication, volume, issue, pages, number of citations, type of document, Digital Object Identifier (DOI).

A comprehensive and detailed search guide can be found in [47]. All defined queries can be url-encoded, saved, and a notification can be enabled to get information about newly published works matching the defined query.

Once the search domain was defined, considering the advanced search functionality of Scopus, four different queries were defined related to the specific EFs among the ones considered: thermal, visual, acoustic, and air quality. In defining the four queries, a screening process was performed before downloading the data to avoid capturing papers not in line with the review’s objective.

The general query structure was the same for the four EFs (Figure 2).

Figure 2 shows how eight concatenated sections characterise the overall structure of the query:The TITLE-ABS-KEY field code (in black) was considered so that only documents in which the selected words appear in the titles, in the abstracts, and in the keywords could be extracted and collected. In particular, the KEY field refers to a combined search that considers, among the other things, the keywords (AUTHKEY) assigned to the document by the author and the controlled vocabulary terms (INDEXTERMS) assigned to the document. Researchers sometimes pay limited attention to the definition of the AUTHKEY, even though keywords play an important role in communicating scientific results [48]. While the INDEXTERMS are added by a team of professional indexers at Scopus, based on different vocabularies [49]. In both cases, when the research area is new or changing rapidly, it is possible to highlight a highly clustered set of new keywords [50].The first set of keywords (in red) was related to the overall topic of wearables, also considering the synonymous acronym PEMS (Personal Environmental Monitoring Systems) or WEMS (Wearable Environmental Monitoring Systems). The operator OR joins the three words such that at least one of the considered terms is included. This Section 2 is the same for all queries that refer to all four EFs.The second set of words (in blue) was connected to the first set of keywords with the operator AND. Section three of the query is characterised by a set of targeted keywords representing each of the four EFs. In particular, Section 3a specifies physical quantities, Section 3b specifies instruments, and Section 3c specifies domains for each EF.Two additional keywords, “environmental” and “monitoring” (in green), were linked to the previous sections with the operator AND. These two terms are separated by the proximity factor “w/3” indicating that they should be within three words of each other without considering the order, thus capturing the simplest form “environmental monitoring” but also more constructed forms such as “monitoring of the environmental (conditions, impacts, etc.)” or “environmental analysis and monitoring”, etc.The “AND NOT” operator (in orange) was used to exclude documents that contain the term specified in the search: “proton exchange membrane” or “fuel cell”, since the acronym PEMS in Section 2 can be misleading because it can refer to a portable environmental monitoring system but also to “proton exchange membranes”, which is often associated with “fuel cell” (the related correct acronym is PEMs with lowercase “s”, but Scopus’ search engine takes into account lemmatisation, that is the process of grouping the inflected forms of a term).The operator OR, associated with the acronym EID, which stands for “Electronic IDentifier” and is a unique alphanumeric string created to identify a record in Scopus, was used to add some other documents, not already considered and derived mainly from the biography screening of both other literature reviews [18,19,20,21,22,23] and the extracted papers derived from Scopus searches with query consisting of Sections 1–5 and 8 of the overall query structure here presented (Figure 2).The operator AND NOT associated with EID was used to exclude some other documents that do not match the research purpose. In particular, Section 7a refers to paper excluded through the screening process performed considering the title and the abstract, while Section 7b refers to manuscripts that were not considered after full-article assessment.The last part of the query (in grey) was used to exclude certain types of documents. In this sense, “cr” stands for “conference review”.

The publication year range was not specified in the overall query structure: it was directly defined by the year of availability of the papers.

Regards the inclusion (and non-inclusion) of some articles via the EID code, as performed in Sections 6 and 7, it is possible to highlight that choosing the potential eligibility or ineligibility of a study, it could be possible to affect the review by a bias [51] due to the personal decision performed by a single author. To avoid this circumstance, a labor-intensive, time-consuming, double data extraction was performed by two authors. As a result, they came to a consensus over discrepancies through discussion and in consultation with the third author, in line with the suggestion reported in [52]. In particular, the EID string provided by Scopus was used in Sections 6 and 7 to assure reproducibility and transparency of the selection criteria, in line with the requirements provided by [53].

Figure 3 reports the Preferred Reporting Items for Systematic reviews and Meta-Analyses (PRISMA) [54] flow diagrams for all four EFs, allowing to present the flow of information on how studies were found, collated, and screened for systematic reviews.

It is possible to highlight how in all four EFs, the additional studies were added only if they were not acquired by Scopus search. Consequently, the duplicates papers are null and the total number of papers considered in the following steps is exactly the sum of the papers from the database search plus the additional ones derived from Section 6.

The final queries for all four EFs are reported in Appendix A.

It can be observed that Section 6 reports only one additional EID for acoustic EF, while Section 7 is quite relevant for air quality EF: this is mainly since some of the research studies report emission measurement monitored by portable systems installed on board of off-road construction equipment, trucks.

For the four queries, the number of selected documents (search performed in April 2021) is of the same order of magnitude for acoustic (11) and visual (16), as well as thermal (19) and air quality (39).

To check the consistency of the considered terms, each of the four .csv was downloaded. Of the various available columns, only those related to the title and the abstract were considered and merged into a single column. The text in each cell of this column was pre-processed: each sentence present in each cell was tokenised, and all punctuations marks, stop words, and words less than three letters in length were removed. Then, an exploratory analysis was performed to check a visual representation of text data using a word cloud defined considering the word cloud package for Python, where the importance of each word is displayed in terms of frequency of occurrence with font size (Figure 4). For each EFs, the 50 most frequent words in the text of the title and abstract were visualised, ensuring that the screening process was performed correctly and that the selected papers were consistent with the aim of the research.

In Figure 4a, the frequent words are light, environmental, sleep, circadian, device, measure, daysimeter, the latter referring to a wearable device used to monitor the light to which the user is exposed.

In Figure 4b, among the most common words are noise, monitoring, exposure, health, wearable, environmental.

In Figure 4c, sensor, wearable, temperature, thermal, urban, and environmental are the most common words.

Finally, in Figure 4d, exposure, monitoring, wearable, sensors, and environmental are the most frequent words.

The document resulting from the search with the four queries were downloaded in “.csv” file format and considered for systematic review.

## 3. Systematic Review

The main findings resulting from the systematic review were defined separately for each of the four EFs.

### 3.1. Monitoring of Visual EF with Wearables

A total of 16 papers were considered for visual EFs, and 12 of them are based on the use of daysimeter, which was first introduced to the scientific community in 2005 [55]. In its first prototypical form, it was equipped with two sensors: a photopic sensor consisting of the Hamamatsu S1223-01 mounted in series with a subtractive glass filter and an opal glass; the other sensor was the Hammamatsu G1962 GaP, mounted in series with a colored glass filter (Schott Glass GG19) to achieve the proper short-wavelength radiation and a gel filter (Roscolux #08, Pale Gold) that provides peak sensitivity to light with a wavelength of 440 nm and is called “blue” sensor for this reason [55]. During the development of this prototype, other alternative sensors were also considered. In all other developments, the device provides the measurement of optical radiation through a glass-filtered silicon photodiode that matches the standard photopic curve and a system consisting of a short-wavelength (blue) sensor equipped with a UV-blocking glass filter that has a long-wavelength cut-off at about 580 nm and a spectral response that peaks at 460 nm. The data obtained from the device are downloaded and post-processed using the human circadian system response to light (CL_A_) model proposed in [56], which is used to define the Circadian Stimulus (CS) [57]. The board to which the photopic sensors are connected is also equipped with two orthogonally aligned accelerometers used to calculate an activity index defined as the root-mean-square deviation of acceleration in two dimensions, monitored every 30 s [57]. In a more advanced version, called Daysimeter-D, the two orthogonally oriented accelerometers have been replaced by three orthogonally oriented solid-state accelerometers to monitor the rest/activity pattern. The Daysimeter-D also allows the measurement of R, G, B, and IR channels with a peak spectral response at 615, 530, 460, and 855 nm, respectively [58]. The device was compared with other similar devices and calibrated in terms of illuminance [lx] and CLA, as reported in [59]. All described alternatives were used for indoor visual applications: laboratory, office, domestic environment, and hospital. As a practical implication, all these studies raise the need to consider a wearable system for measuring optical radiation in many applications related to human well-being, which may provide important insights into the relationship between circadian disruption [57] and well-being [60,61,62,63,64,65,66,67].

In other studies where the daysimiter is not considered, the scope of the research are different: in [68], a miniaturised microclimate station, specifically tailored to be worn while walking or cycling and therefore collecting data according to the pedestrian perspective in anthropogenic areas, was used to collect data on illuminance levels and some other variables related to thermal and air quality EFs; in [69], a calibrated wearable device, called Eco-Mini, used for environmental monitoring is presented; in [70], the authors showed how a textile composed of Janus chromic fibers shows stable practical performance and excellent sensitivity to UV/IR radiation to achieve real-time, energy-free, visual monitoring of IR radiation temperature and UV index; finally, in [71], the authors presented a fully low-cost and open-source (in accordance with the requirements defined by UNICEF, see Section 1.1) wearable light data logger for studying physiological and psychological effects of light.

Appendix B, Table A1, lists the papers that were selected and considered for this particular EF.

### 3.2. Monitoring of Acoustic EF with Wearables

A total of 11 papers were considered for acoustic EF.

In [72], the authors presented a wrist-worn device consisting of four layers: the two outermost layers are used to monitor the sound level and gasses in the environment. The two sensor layers are connected to the host layer by a layer called “flex interface” by the authors, which allows the connection of new hardware without developing a new physical layer. The main objective of this device development is to measure personal exposure to several physical (air humidity and temperature and air pressure) and chemical environmental parameters (CO and NO_2_) known to be hazardous, in addition to the sound pressure level monitored by an analogue sound sensor MLMS-EMGN-4.0. In [73,74], the same authors focused on the design and development of the case of this wearable, which was manufactured using additive manufacturing techniques. In [75], the same authors reported the development of this multi-layered wearable that can capture parameters from the environmental, behavioural, and physiological domains.

In [76], the researchers presented the NEATVIBEwear device developed for the personal measurement of exposure to ultrafine particles (UFP) and noise in a pediatric population (it was tested at home and school), allowing the independent and/or combined effect of these environmental health exposures to be studied.

In [77], the authors justify using a smartphone with the NoiseSpy application and the Empatica E4 wearable with a striking correlation between noise and heart variability in outdoor environments. Particular is the case of [78], where workers employed in molding and in artifacts finishing were equipped with a commercial wearable device (Quest DLX-1) to monitor noise exposure. Biological monitoring of styrene exposure using urine concentration was also conducted. The study revealed that the group of workers exposed to high noise level was also exposed to low styrene concentration. The group exposed to high styrene concentration near the limit of 20 ppm was also exposed to the lowest noise level. This resulted in a significant negative correlation between otoacoustic emission and styrene concentration. In [69], the calibrated portable device Eco-Mini, already mentioned in the section of Visual EF, also allowed the measurement of Sound Pressure Levels. In [79], the authors demonstrated the feasibility of collecting personal Particulate Matter (PM2.5), language, noise data, cognitive assessments, and biospecimens from a sample of 3-4-year-old children using wearable ultrafine particle sensors and LENA—Language Environment Analysis System. Two different approaches were then collected to provide an example of real-time information on urban acoustic pollution [80] and overall personal pollution [81], where the PONG device can monitor ambient sound levels as well as VOCs and NO_2_, UVA and UVB, air temperature, relative humidity and air pressure data.

Appendix B, Table A2, lists the papers that were selected and considered for this particular EF.

### 3.3. Monitoring of Thermal EF with Wearables

A total of 19 papers were considered for thermal EF.

The studies reported in [68,81] have already been presented and refer to the possibility of collecting thermal data from the pedestrian perspective in the city. In [82], the wearable enables the acquisition of representative data of urban microclimate conditions at a pedestrian level during a heatwave in a historical hilly city in central Italy. These data can be used to investigate microclimate variations within the city due to urban configuration and architectural design, human activities, and anthropogenic actions responsible for local overheating and to calculate direct thermal indices for human comfort assessment. In [83], the authors reported a research study on multi-stimuli (air temperature and UV) responsive chromism capable of displaying different colours in four different temperature ranges (blue at T < 15 °C, green at 15 °C < T < 33 °C, red at 33 °C < T < 65 °C, and white T > 65 °C) covering a wide range of applications, more than just monitoring environmental conditions. In [84], the authors used different wearables to evaluate the effects of hair exposomes in Brazil. In [85], the authors estimated Heat Exposure of public service workers in Birmingham, Alabama, using thermometers attached to the workers’ shoes. In [86], the authors present We-Safe, a self-powered sensor network system used for safety applications. In [87], the authors investigate an artificial, low-cost, skin-like temperature sensor that was highly flexible and provided for visual evaluation of temperature based on pectin and xanthan gum, which can be used in many fields such as electronic skin, human body temperature measurement, and environmental monitoring. Thus, this sensor can be used to alert people in real-time to prevent health issues resulting from extreme changes in human body temperature. In [88], the authors demonstrated the feasibility of a wearable wireless sensor system that can be attached to a uniform and used for temperature monitoring and can be activated remotely by an RF control signal. In [89], the authors presented a novel MoS_2_/Cu_2_S hybrid grown on disposable cellulose paper using a hydrothermal method to measure relative humidity and temperature, among other physical quantities. The data can be wirelessly transmitted to a smartphone with an appropriate application. In [90], the authors presented a wearable sensor for analysing personal exposure to the thermal environment and air quality (ozone, particulate matter, CO). The potential of the proposed solution is to fill the gap left by traditional air pollution monitoring. However, it still has a limitation due to the power consumption of the sensor. In [91], the authors present a smart indoor environment monitoring system for safety applications. It is based on custom wearable sensor nodes connected to a static WSN. The system is designed for a hazardous gas environment and thermal monitoring. However, it could also be used for several other safety applications or other areas, such as tracking medical devices in a hospital. In [92], a wearable system for continuous environment and health monitoring in chronic respiratory disease is presented, consisting of a wristband, a chest patch, and a handheld spirometer. In [93], the authors developed “MyPart” to address the need for a wearable device that combines accuracy, low cost, and portability in a single design. The study also reports the results of a preliminary user study conducted to evaluate the overall system’s experience with appropriate results in terms of overall performance. In [94], various subjects wore a series of instruments that recorded individual microclimatic and physiological responses along a fixed pedestrian route that passed through various urban structures. Subjects experienced different thermal environments that could not be represented by fixed-point routine observation data. A clear dependence of sweating on gender and body size was found; men sweated more than women; overweight subjects sweated more than standard/underweight subjects. The temperature of the skin (Tskin) had a linear relationship with Standard Effective Temperature and a similar clear dependence on sex and body size differences: Tskin of the higher sweating groups was lower than that of the lower sweating groups, reflecting differences in evaporative cooling by sweating. In [95], the authors report on the design and initial deployment of the Citisense mobile air quality sensing system, which collects information about the thermal environmental and ambient air quality. In [96], the authors present in-depth knowledge on sensor selection and calibration of sensors for wearable applications for thermal and air quality assessment. In [97], laboratory testing of the prototype UPAS (Ultrasonic Personal Aerosol Sampler) shows excellent agreement with equivalent samplers in the federal reference method for gravimetric analysis of PM_2.5_ over a wide range of concentrations. UPAS also monitors thermal variables (air temperature and relative humidity). In [98], thermal data of the environment and skin temperature are collected by wearable sensors and a cloud platform for monitoring environmental parameters in e-health applications. The authors suggested that this type of monitoring system is not suitable for critical situations. However, many people suffering from chronic diseases and their families can benefit from this type of system.

Appendix B, Table A3, lists the papers that were selected and considered for this particular EF.

### 3.4. Monitoring of Air Quality EF with Wearables

A total of 39 papers were selected for air quality EF.

Depending on the substance to be monitored, all these studies can be divided into four main categories. The first group consists of research studies focusing on monitoring Volatile Organic Compounds (VOCs). VOCs include an extensive family of chemical substances: from aromatic to ketones, through alcohols, aliphatic, hydrocarbons, aldehydes, ethers, and acids. They can be emitted by building materials and during cooking, and increase their concentration indoors if they are poorly ventilated [99]. On the other hand, in outdoor spaces, the primary sources of VOCs are roads traffic, fossil fuel combustion, and pesticides. Due to the vast differences among chemical substances, the effects on human health from exposure to VOCs can be diverse: from skin or eye irritation to headaches, nausea, and cancer [100]. The selected studies can be divided into those in which wearables were used only for outdoor monitoring in this category. In particular, in [101,102], the wearable “Microfabricated Preconcentrator Chip—μPC” was used to show the potential applications in occupational risk assessment for specific occupations, such as industries involving direct handling of petroleum products or in large-scale asthma population studies in pediatric and teenagers. In [68], the previously presented wearable can also measure pedestrian exposure to VOCs in cities. In [103], the authors presented WearAir, an original expressive T-shirt that detects the air quality of the wearer’s environment based on the measured volatile organic compounds. WearAir may be useful in motivating others to explore ways to communicate environmental information to lay people more effectively. The second subcategory refers to the study in which wearables were used indistinctly in indoor or outdoor environments. In [104], the authors demonstrated with a practical application the potential of the proposed wearable device to cope with many real-world analyte monitoring applications.

In [105,106], the authors presented a cost-effective and reliable platform for personal exposure assessment. Several comparisons and tests showed that the proposed VOC device is suitable to characterise personal exposure in many real-world scenarios.

In [107], the authors presented a wrist-worn Asthma Research Tool (ART) designed to identify and detect asthma triggers using only low-cost components.

In [69], a wearable called “Eco-Mini” was presented, which the authors concluded overcomes the limitations of the first generation of low-cost environmental monitors that were generally not suitable for clinical environmental health studies due to practical challenges such as calibration, reproducibility, form factor, and battery life. In [108], the proposed wearable was able to detect not only VOCs but also O_3_, NO_x_, and CO_x_.

The second group of research studies consists of those that focus on monitoring Particulate Matter (PM), also called particle pollution. PMs come in many sizes and shapes and can consist of hundreds of different chemicals. Some are emitted directly from sources, such as construction sites, dirt roads, fields, smokestacks, or fires. In contrast, others result from the atmospheric reaction of SO_2_ and NO_2_, which in turn are produced by power plants, industries, and motor vehicles [109]. This research study category can be divided into those where the corresponding wearables are mainly used indoors, and those where they are indiscriminately applied indoors or outdoors, and those that are mainly outdoors. Research studies related to the first case include [110], where a wearable called “Ultrasonic Personal Aerosol Sampler—UPAS” was used to monitor PM_2.5_ in households in Honduras where rural women use wood-fired cookstoves. In [111], the authors conducted a research study to assess the exposure of worshippers to PM_10_ and PM_2.5_ in two different types of Buddhist temples in Tai-Chung. Samples were collected using Personal Environment Monitors (PEMs) connected to personal pumps flow rates of 2 L/min. The PEMs were worn by research staff, who mimicked the activities of the worshippers to determine their exposures to PM_10_ and PM_2.5_ in both temples. To reduce exposure to PM_10_ and PM_2.5_, the authors recommend spending less time in Buddhist temples, choosing a well-ventilated temple, or avoiding visiting temples on the first and 15th days of the lunar month. In the previously presented study [76], the authors used wearables to monitor noise and assess personal exposure of UFP (ultrafine particles) in the pediatric population, allowing researchers to examine the independent and/or combined effects of these health-related environmental exposures. The same research objective is pursued in the previously presented work [79], in which, in addition to the thermal and acoustic aspect, wearables worn by 3-4-year-old children are used to collect PM_2.5_ and PM_10_ data at school and home, demonstrating that it is possible to collect personal Particulate Matter with wearables when considering this population. In the previously presented work [90], the EnviroSensor 2.0 wearable was used in addition to thermal and location data in a laboratory test to evaluate the potential of this device to collect particulate matter, ozone, and CO concentration data.

Considering the studies on indoors and outdoors PM monitoring, the authors in [112] demonstrated the feasibility of using wearables for real-time remote air quality and health monitoring applications. In [113], an experiment was conducted in Athens-Greece to collect PM_1_, PM_2.5_, and PM_10_ data. It was shown that residential air quality was determined by the type and intensity of outdoor air sources and their vertical distance from the street. Indoor activities such as cooking and cleaning further increased PM concentrations and formulated air quality, while particle accumulation was evident. In [114], personal pollution monitoring and GPS tracking were used to assess children’s PM_2.5_ exposure in their everyday environment. The overall results of this study indicate that mean PM_2.5_ exposure was lowest for children who walked to and from school and higher for those who were driven. In [115,116], the authors presented the results of a study conducted under the European Research Council funded “Cardiovascular Health effects of Air pollution in Telangana, India—CHAI” project, which investigated the association of particulate air pollution from outdoor and household sources with markers of atherosclerosis. In the previously presented study [93], the wearable “My part” was developed, distinguishing and counting particles of different sizes.

Finally, only two studies monitored PM only outdoors. In [117], a correlation was found between air quality indicators, participants’ subjective feelings about air quality, physical activity status measured with wearable sensors, and reported health symptoms. Of particular note is the result of [118]. The authors used data collected by wearables to develop a machine learning model to identify periods of cycling activity necessary for estimating the inhaled dose of chemicals.

The third group of research studies consists of those that focus on monitoring of the NO_2_ and CO. The first chemical is responsible for respiratory irritation or asthma in case of prolonged exposure [119]. While the second one, especially at elevated outdoor concentrations, may be of particular concern to people with some types of heart disease [120]. All the collected studies [45,72,75,95,121,122,123] explain the development phase, the practical test and validation of a customized wearable device for environmental monitoring.

The fourth group refers to studies focused on monitoring of CO_2_ concentration with simple wearables [124] or wearables exchanging data with a Wireless Sensor Network (WSN) [91,125] or a LoRa network [126].

Finally, among the collected research papers, five cannot be classified into the above-reported groups. In [88], the authors demonstrated a wearable wireless sensor system that can be attached to a uniform and used to monitor the combustible gas concentration and air temperature that a RF control signal can remotely activate. In [127], a wireless wearable ring-based sensor system for rapid electrochemical monitoring of explosives and nerve agent in vapor and liquid phases was presented. In [92], the wearable collects ozone concentration data among the other thermal, biometric, and activity data. It is interesting to note the case of study [84] where two wearables, the MyExposome wristband and the 2BTech Personal Ozone Monitor, were used to collect data on environmental aggressors (Polycyclic aromatic hydrocarbons—PAHs, oxygenated PAHs—OPAHs, Polychlorinated biphenyls—PCBs, Pesticides, organophophorous flame retardants—OPFRs, Surface ozone content—SOC) associated with hair damage. In [78], a significant negative correlation was found between the otoacoustic emission levels and the concentration of the styrene urinary metabolites, in workers employed in molding and in artifacts refining.

Appendix B, Table A4, lists the papers that were selected and considered for this particular EF.

## 4. Discussion and Conclusions

Considering the Scopus functionality for combining search queries, a total of 68 papers can be summarised by entering the list number of each stored search query and the operator OR, some of which cover more than one EF. Figure 5 shows the annual scientific production for each EF and all 4EFs. In this last case, the papers that consider wearables used for multiple EFs are counted only once. The period of publication is between 2000 and 2021. The first paper published in 2000 is related to the air quality research study conducted to assess the exposure of worshippers to PM_10_ and PM_2.5_ in two different types of Buddhist temples in Tai-Chung [111]. The second one, published in 2005, refers to visual EF when the daysimeter was first introduced to the scientific community [55].

Even though the systematic review covers the last 20 years, it can be highlighted from Figure 5 how only from 2012 onwards the most significant increase in production was recorded, with an annual growth rate of 12.18% considering all 4EFs.

Of the 68 papers, most consider wearables that can monitor various EFs. The wearables used to assess visual aspects focused mainly on indoor environments. Several studies looked at the impact of daylight on circadian rhythms.

As mentioned above, wearables used for visual EF were mainly focused on using daysimeters in different contexts or, alternatively, in defining the UV, IR, or illuminance values. Probably, in the future, studying the feasibility of using some type of wearable spectrometer for visual EF assessment could be considered, allowing the effects of visual aspects in the built environment to be examined from a human-centered perspective, more than has already been done.

Acoustic EF studies have mainly focused on investigating the health effects of combined acoustic and environmental air exposures. In a case study [77], a smartphone was used as a wearable monitoring device to understand the influence of noise on heart rate variability. The main problem in this perspective is accuracy across devices: different devices will have different microphones with different sensitivity levels, making measurements vary from device to device [128]. However, a more combined integration between physical wearables equipped with different sensors and a smartphone-based application also developed using an open-source framework [129], and integrated hardware could be interesting.

While for thermal EF, the use of wearables to assess outdoor comfort could be considered widespread, for example, tying the use of these wearables to specific outdoor conditions such as the Urban Heat Island and associated impacts on human well-being; instead, it is usually used in combination with some others to determine the overall environmental conditions while conducting the test.

Undoubtedly, among the four considered EFs, air quality is the most considered in the use of wearables, and the case studies span multiple domains, as seen in Section 3.4.

As the presented review shows, the data collected by wearables can be used to define the spatio-temporal propagation of environmental parameters. In some cases, these data are combined with subjective biometric data to identify a potential pattern. In a small number of cases, the environmental and biometric data are combined with subjective feedback. This aspect could probably be explored more deeply in the near future, in a human-centric approach. As mentioned in introduction, wearables have also been used to acquire data and assess physical ergonomic risk factors, particularly on unfavourable postures as reported in [13] where it was focused on the design principles of wearables from an ergonomic perspective as one of the main future directions that this type of device could address.

In healthcare monitoring [14], wearables have been used in research studies for monitoring chronic diseases such as asthma, cardiovascular diseases, diabetes and nutrition, gait and fall, neurological diseases, stress. In [15], it was pointed out that wearable devices used for health monitoring will face several limitations due to practical difficulties in achieving user-friendly solutions, security and privacy issues, and lack of industry standards.

In [16], it was emphasised that, although a considerable research effort has been devoted to the benefits of wearables in the work environment, less attention has been paid to the empirical analysis of employees’ acceptance of wearable technology.

Ergonomic design principles, user-friendly solutions, and user acceptance of wearable technology are three main aspects that can be highlighted as the major directions that wearables used for environmental monitoring will also need to consider, more than it already has been done. From the table in Appendix B, there are no clear indications of the limitations of the proposed devices. However, after reading and viewing the photos reported in the papers, these wearables could be likely to be better integrated with clothing and apparel.

In 60% of the total cases, the accuracy of the tested system was performed simply by correlating monitored data with those acquired with calibrated sensors of a reference instrument. The percentages are quite different if each EF is considered separately: In 87% of the cases, a comparison or a calibration was performed for wearables used for monitoring visual aspects, while the accuracy of wearables used in the acoustic and thermal domain is verified in 45% of the cases. Even lower is the case of air quality monitoring using wearables, with only 40% of the tested devices. This is a limitation of the proposed wearables because a test of accuracy, if done consistently and in all cases, may reduce much of the variability and limitations of not-calibrated hardware. For all the four EFs, the devices are generally characterized by great usability when using the hardware while performing the monitoring campaign. The usability of software used for data extraction and analysis is not contemplated because it is performed in all the cases by specified researchers and not by direct involvement of participants who generally are not even aware of the measured value of the general environmental variable.

There is one point that did not emerge in this review paper that we would like to highlight as food for thought for likely future studies: related to one of UNICEF’s requirements for this type of device (Section 1.1): of the 68 studies, how many fully shared their wearable to encourage dissemination by releasing it in open-source form? The answer is as disappointing as ever: only one. Work should be done in this direction, encouraging the proliferation of open-source wearables, to promote greater scalability and the proliferation of several devices applicable in different domains. In this sense, countries that are distinguished for their technologically advanced economy and society are reconsidering their education policy and are transferring funding to promote STEM education, an acronym that integrates the academic disciplines of Science, Technology, Engineering, and Mathematics [130], and probably, considering the investigated topic of wearables for environmental monitoring, especially when released under an open-source license, it could encourage both the development of more environmentally conscious society on the one hand and the spread of coding and programming in schools on the other [131].

Following the same principles of dissemination of information in an open and shared way, we decided to publish in an open-source journal, despite not having received funding for the research carried out. We have also shared the queries used in this systematic review (see Appendix A) to verify, in the future, how the studies in this specific area of research will evolve.

This is a future and distinct research direction for improving a widespread wearable device. We hope that this systematic review will highlight the current state of innovative work in wearable natural checking and provide rules to improve further the design and development of this type of device and related practical applications.

## Figures and Tables

**Figure 1 sensors-21-04727-f001:**
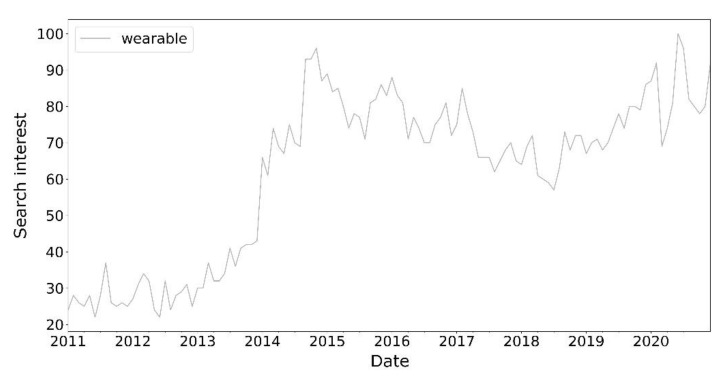
The global trend over the last 10 years of the keyword “wearable”—Google, 2020.

**Figure 2 sensors-21-04727-f002:**
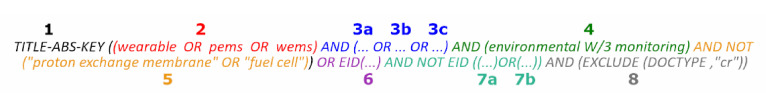
Query structure consisting of eight main sections.

**Figure 3 sensors-21-04727-f003:**
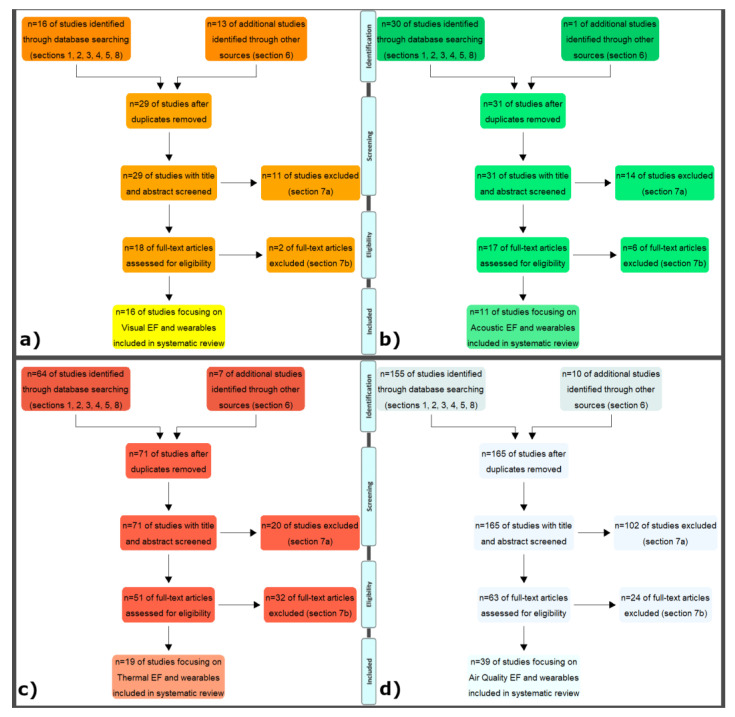
PRISMA flow diagrams: (**a**) Visual EF; (**b**) Acoustic EF; (**c**) Thermal EF; (**d**) Air Quality EF. The number of sections refers to query structure (Figure 2).

**Figure 4 sensors-21-04727-f004:**
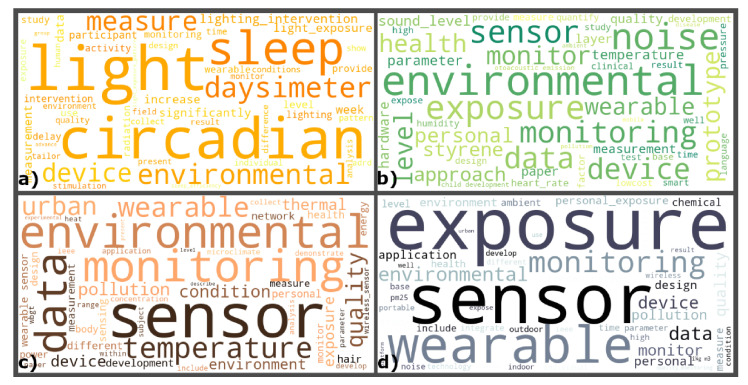
Exploratory Analysis with the word cloud of the most used terms in the title and abstract of the selected papers: (**a**) Visual EF; (**b**) Acoustic EF; (**c**) Thermal EF; (**d**) Air Quality EF.

**Figure 5 sensors-21-04727-f005:**
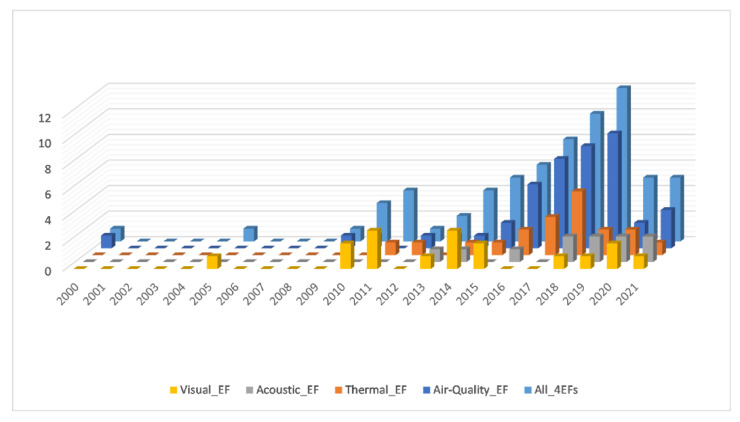
Annual Scientific production—Scopus, 2021.

## Data Availability

Not applicable.

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
