# Peer review of "Wearable Devices for Environmental Monitoring in the Built Environment: A Systematic Review"

_sensors, 2021, doi:10.3390/s21144727_

Round 1

Reviewer 1 Report

This paper conducts a systematic review on wearable devices for the four different environmental factors monitoring (visual, acoustic, thermal and air quality) and analyzes the research trends of wearables for environmental monitoring. Here are some kind comments for this article.

  1. It is better to provide some pictures about wearable monitoring devices, which will help readers to understand easily.
  2. In page 4, the concept of personal environmental monitoring system (PEMS) is first introduced. The definition of this concept and its relationship to the concept of wearable environmental monitoring system (WEMS) should be explained.
  3. The section of discussion and conclusion should be further analyzed and summarized. More technical issues such as the accuracy and usability of wearable environmental monitoring system need to be discussed.

Author Response

The authors wish to thank the reviewer for providing constructive feedback. Please see the attached file for responses to your comments highlighted in yellow point by point. The authors tried to address those to the extent possible.

Reviewer 2 Report

On page14, in lines 618 and 634,  "from 2005 to 20021,  from 2013 to 20021", should be "from 2005 to 2021,  from 2013 to 2021".

Author Response

The authors wish to thank the reviewer for providing constructive feedback.  

We modified the two typos in the two lines (now 705 and 721) of the revised manuscript.

Reviewer 3 Report

In the manuscript ID: sensors-1275713, entitled "Wearable devices for environmental monitoring in the built environment: a systematic review" the authors try, through a systematic review of the literature, to understand the trend in the scientific literature regarding the use of wearable instrumentation. The manuscript is certainly of interest and worthy of publication although I suggest some revisions before publication. My major concerns regard the discussions: the authors stated that they will answer this question: "What are the main areas of interest covered by scientific production?": although this issue has been well described and detailed, I believe there has been an a priori selection of these areas of interest. I therefore believe that a discussion of other possible areas not analyzed by the authors is necessary. In addition, I believe that the discussions can be better focused on the limits and advantages that this instrumentation can bring in the different fields of application identified by the authors. Finally, my major concern regards the inclusion (and non-inclusion) of some articles via the EID code: in my opinion, this can affect the systematic review process, because the authors arbitrarily choose which articles to include in the review of the literature. Minor comments are reported below:

  • Lines 20-27: I believe that this part of description of the methods used is of little use in the abstract and I would reduce it to one line, leaving space for the main results of the study (which in any case I suggest inserting here).
  • Figure 1: Although intuitive, I suggest adding the definition of the axes into the chart.
  • Line 157: Even if this concept can be shared (“[…] was used due to the high quality of the available resources […]”) I suggest to the authors to be as objective as possible.
  • 3. Database for research purpose and metadata: I think this paragraph is not very useful: it only describes the database used for the systematic review and I really think it is not very interesting, for the purposes of the study. If the authors deem it appropriate to keep this paragraph, however, I suggest moving it to the “methodology” section.
  • Line 186: I suggest authors be more specific about the process of exclusion of the different papers, basing (and maybe even inserting in the text) on the PRISMA criteria (see above all the flowchart).
  • Description of figure 2: Although I think this figure is a great expedition to explain how the 4 search strings were defined, I suggest being more schematic in the description of the figure (line a194). For example, I do not believe that the terms AUTHKEY were used in the figure and for this reason I believe that such a detailed description of the structure and operation of the search string is of little use in this context.
  • Line 242: When was the last database search performed?
  • Lines 243-261 and 275-279: I think this part is too verbose and can be reduced.
  • Figure 3: Congratulations to the authors!
  • Paragraph 2 "Methodology": I advise the authors to better define in this paragraph the inclusion and exclusion criteria used for the selection of the papers.
  • Line 548: It is not clear whether a priori temporal exclusion criteria have been included in the selection of articles. I suggest specifying it better in the method description. In addition, why are different time frames shown in Appendix A?
  • Line 589: I believe that the limitations (as well as the advantages) associated with this instrumentation can be better discussed and explored by the authors.
  • Appendix A: My major concern regards the inclusion (and non-inclusion) of some articles via the EID code: in my opinion, this can affect the systematic review process when authors arbitrarily choose which articles to include in a review of the literature.
  • Appendix B: I suggest that authors review this table, so that only the information that is most important for the purpose of the review is reported. In addition, why insert only the title of the article in the question and not all references?
  • Beware of some minor typos (example: PM10 as subscript).

Author Response

(The authors gave the same response as above.)
